# Uncovering overlooked bacterial infections in gene X-pert MTB/RIF negative sputum samples of adult patients with presumptive tuberculosis in Harar, Eastern Ethiopia

**Negesso Gebre[1], Jemal Mohammed[2], Rudwan Yasin Abrahim[3], Mohammed Ahmed[2]\*, Kedir Urgesa[2]**

**1** Higher Health Center, Haramaya University, Haramaya, Oromia, Ethiopia, **2** School of Medical Laboratory Sciences, College of Health and Medical Sciences, Haramaya University, Harar, Ethiopia, **3** School of Medicine, College of Health and Medical Sciences, Haramaya University, Harar, Ethiopia

\* mameahmed129@gmail.com

## Abstract

### Background

Lower respiratory tract infections are among the diseases that pose an existential threat to global public health. Clinical presentations of various bacterial lower respiratory infections and presumptive pulmonary tuberculosis are often overlap, hindering prompt patient care and correct diagnosis. In developing countries like Ethiopia, the management of presumptive tuberculosis cases that test negative for Gene Xpert is often empirical. Consequently, it is imperative to address potential overlooked bacterial infections in cases that are negative for tuberculosis.

### Objective

This study aimed to determine the prevalence of pathogenic bacteria, antimicrobial susceptibility patterns, and associated factors of lower respiratory tract infection among Gene Xpert-negative presumptive tuberculosis adult patients at Hiwot Fana Comprehensive Specialized Hospital, eastern Ethiopia.

### Method

A hospital-based cross-sectional study was conducted among 371 study participants from January 01 to April 30, 2024. Socio-demographic and health related data were collected using structured questionnaire. Gene Xpert MTB/RIF was used for initial tuberculosis screening, and sputum culture was used to isolate bacterial pathogens. Bacterial isolates were identified based on gram staining, colony characteristics, and biochemical reactions. Antimicrobial susceptibility test was done using Kirby-Bauer disc diffusion method. Methicillin resistance *S. aureus* was confirmed using cefoxitin

**Data availability statement:** All relevant data are within the paper and its Supporting Information files.

**Funding:** The author(s) received no specific funding for this work.

**Competing interests:** The authors have declared that no competing interest exists.

(30 µg). Logistic regression analysis was used to assess the association between outcome and predictor variables.

## Results

The overall prevalence of pathogenic bacterial lower respiratory tract infection was 34.0% (95% CI; 29.2, 38.8). Gram-negative bacteria accounted for 66.7%. *S. aureus* (19.0%) was the predominant isolate followed by *K. pneumoniae* (14.3%). The proportion of multi-drug resistant bacteria and methicillin-resistance *S. aureus* was 39.7% and 16.7% respectively. Primary education (AOR = 4.01; 95% CI = 1.62, 9.92), history of antibiotic usage (AOR = 1.87; 95% CI = 1.13, 3.08) and crowded living condition (AOR = 5.11; 95% CI = 3.03, 8.61) were factors associated with pathogenic bacterial lower respiratory tract infections.

## Conclusion

This study revealed that patients with presumptive tuberculosis who tested negative on Gene Xpert- were predominantly infected with gram negative pathogenic bacteria. Therefore, it's important to prioritize sputum culture and antibiotic susceptibility testing. This study underscore the need to avoid the misuse of antibiotics and crowded living conditions.

## Introduction

Globally, lower respiratory tract infections (LRTIs) are a leading causes of morbidity and mortality, particularly in developing countries [1,2]. Global burden of disease study 2019 indicates that, in the last three decades LRTIs causes an estimated 489 million incidence cases and 2.4 million deaths of which Sab-Saharan Africa regions accounts largest incidence and mortality from lower respiratory infections [3]. In Ethiopia, LRTIs are the main reason for hospital admissions and the third leading cause of death, accounting 8.2% of all deaths reported in 2019 [4]. In cases presumptive tuberculosis that test negative using Gene Xpert, the most frequently identified bacteria responsible for LRTIs includes *Klebsiella pneumoniae*, *Staphylococcus aureus*, *Haemophilus influenzae*, and *Streptococcus pneumoniae* [5,6].

According to the endorsement by WHO (2017), the Gene Xpert MTB/RIF assay is highly specific (98%) to give an excellent performance for the diagnosis of TB [7]. Thus, patients with Gene Xpert MTB/RIF negative are unlikely to suffer from pulmonary tuberculosis [8]. However, LRTIs that have similar clinical manifestation (like cough, fever, chest pain and difficult breathing) with presumptive TB are polymicrobial in nature including (bacteria (*S. pneumoniae, K. pneumoniae*), fungi (Aspergillus), viruses (Influenza, respiratory syncytial virus), and parasites some endemic areas) and rendering emergency of Antimicrobial resistance (AMR) [9–11].

Antimicrobial resistance (AMR) by pathogenic bacteria is dramatically accelerated by exploiting antibiotics in the communities and health facilities, leading to the

emergence of resistant bacteria [12]. Methicillin Resistant *Staphylococcus aureus* (MRSA) is one of the emerging multi-drug resistance strain circulating in health facilities and communities [13,14]. Bacterial LRTI pathogens are associated with different factors including old age, crowded living conditions, smoking, comorbidities, and history of prior antibiotic usage [15,16].

According to Ethiopian national guidelines for management of TB, DR-TB and leprosy by Ethiopian Federal Ministry of Health (2018), patients with respiratory symptoms with whom tuberculosis (TB) is less likely are empirical [17]. However, Gene Xpert MTB/RIF negative presumptive pulmonary TB cases are predominantly infected with other pathogenic bacteria than TB [18], which could result in poor patient's prognosis and prone to emergence of multi-drug resistant bacterial pathogens [19]. Therefore, it is crucial to consider other bacterial pathogens causing LRTIs while implementing TB diagnostic protocols to ensure accurate diagnosis and optimal patient care [20].

Although few studies have been done in Ethiopia on bacterial LRTIs [5,20–23], study on the pathogenic bacteria lower respiratory infection, antimicrobial susceptibility patterns and associated factors among presumptive tuberculosis adult patients with Gene Xpert-negative is rare, especially in eastern Ethiopia there is no published data.

Thus, this study aimed to determine the prevalence of pathogenic bacteria, antimicrobial susceptibility patterns and associated factors of lower respiratory tract infections among Gene Xpert-negative presumptive pulmonary tuberculosis adult patients at Hiwot Fana Comprehensive Specialized Hospital, Harar, eastern Ethiopia.

## Materials and methods

### Study area and period

The study was conducted at Hiwot Fana Comprehensive Specialized Hospital (HFCSH) in Harar, Harari region, eastern Ethiopia from January 01 to April 30, 2024. Harar, the capital city of Harari region, is located approximately 525 km east of Addis Ababa. The hospital serving about 5 million people in Harari regions, Dire Dawa, Eastern and West Hararghe, Somali region as well as a clients from neighbouring countries. The hospital provides the general public with complete laboratory services including Gene Xpert MTB/RIF assay for presumptive tuberculosis diagnosis. Based on the hospital's annual service report, approximately 3,500 individuals were identified as presumptive tuberculosis cases during the study period.

### Study design, population, sampling technique and procedure

An institutional-based cross-sectional study was conducted. Presumptive pulmonary tuberculosis patients from both outpatient and inpatient department who visited laboratory for Gene Xpert MTB/RIF assay services and adult patients aged ≥ 18 years with Gene Xpert-negative who consented were included in the study. Convenient sampling method were used to select study participants until the calculated sample was fulfilled.

Adult patients who had received antibiotic within two weeks prior to the study period, those who were critically ill, and individuals unable to provide sputum samples were excluded from the study.

### Sample size determination

The sample size was determined using a single population proportion formula by taking the prevalence (33.5%) from a previous study done in Hawassa [21]. Considering the 95% confidence interval CI (z = 1.96), margin error (5%), (d = 0.05). The calculated sample size was 342 Therefore, final sample size was 376 after adding 10% non-respondents.

### Operational definition

Lower respiratory tract infections are defined as any infections in the lower respiratory tract with common symptoms like cough, fever, shortness of breath, chest pain or tightness, which include pneumonia, bronchitis, and bronchiolitis [2].

Gene Xpert MTB/RIF-negative presumptive pulmonary TB cases: refers to cases where Gene Xpert MTB/RIF assay does not detect *Mycobacterium tuberculosis* and rifampicin resistance in patients suspected with presumptive pulmonary TB [5].

Crowded living condition: refers to if there are more than three people per habitable room or more than two people per bedroom [24].

## Data collection methods

Data were collected by two trained professional nurses and one medical laboratory technologist. All relevant data, including socio-demographic variables, behavioral factors, health-related conditions, and environmental factors, was collected using a pre-tested structured questionnaire adapted from different literatures [5,18,20,21,25,26].

## Sputum samples collection, and Bacterial identification method

Purulent sputum sample were collected into a wide mouthed screw plastic cup from all participants. Prior to sputum collection participants were briefly instructed to rinse their mouth with water and take deep breath to expectorate purulent sputum. The quality of collected sputum samples were assessed using Bartlett's criteria through the score of pus cells, and squamous epithelial cells on gram stain. Based on gram stain, the sputum samples with greater than 25 polymorphonuclear and less than 10 epithelial cells per low power field were accepted for culture and transported immediately to microbiology laboratory within less than 2 hours of collection. Then, Sputum samples were inoculated onto Blood agar (Oxoid Ltd., Basingstoke, UK), Chocolate agar (Oxoid Ltd., Basingstoke, UK) and MacConkey agar (Oxoid Ltd., Basingstoke, UK) by using sterile wire loop. Briefly, MacConkey agar (Oxoid Ltd., Basingstoke, UK) plates were incubated aerobically at 37 °C for 24 hours, whereas Blood and Chocolate agar plates were incubated within 5% $CO_2$ generating candle jar at 37 °C for 24 hours. The plates were inspected for the growth after 24 hours of incubations. The pure isolates of bacterial pathogens were characterized by colony morphology, gram stain, and hemolytic reaction. Identification of pathogenic bacteria down to species level was performed using standard biochemical tests [27].

## Antimicrobial susceptibility testing

Antimicrobial susceptibility test (AST) was performed using a modified Kirby-Bauer disk diffusion technique as recommended by Clinical Laboratory Standard Institute Guideline stated in CLSI (2022). Suspension of isolated bacteria was done by using normal saline 0.9% to emulsify colonies until matching to 0.5% McFarland standard solution. Using a sterile cotton swab, sufficient inoculum was distributed on Muller-Hinton agar plates with or without 5% sheep blood with (Oxoid Ltd., Basingstoke, UK). *H. influenzae* media (HIM) was used to perform AST for H. *influenza* as stated in *CLSI* (2022). After 5 minutes, a set of selected antimicrobial disks were aseptically placed on Mueller-Hinton agar plates and allowed to stand at room temperature for 15 minutes. Then, all plates were incubated at $37^0$C for 24 hours while maintaining all the requirements of the respective bacteria as done during the isolation process. The diameters of the zone of inhibition around the disk was measured using a ruler and compared to reference points stated in CLSI (2022) and interpreted as sensitive (S), intermediate (I), and resistant (R) [28]. Isolates were classified as either susceptible or resistant to an antibiotic and all isolates with intermediate resistance were classified as resistant for a better fit in further statistical analysis. Multi-drug resistance (MDR) in this study was extrapolated as resistance to three or more groups of antibiotics tested [29]. Isolates of Methicillin *S. aureus (MRSA)* were detected according to the CLSI guidelines by using cefoxitin disc. Fifteen antibiotics were selected based on CLSI (2022) recommendation, guideline and routine local antibiotic usage, which include Ampicillin (10 µg), Amoxicillin-clavunate (30 µg), Penicillin (10 µg), Piperacillin/tazobactam(100 µg), Ceftriaxone (30 µg), Ceftazidime (30 µg), Cefoxitin (30 µg), Gentamicin (10 µg), Erythromycin (15 µg), Azithromycin(15 µg), Cotrimoxazole) (12 µg),Tetracycline (30 µg),clindamycin (2 µg), Ciprofloxacin(5 µg) and Meropenem (20 µg).

## Data quality control

Prior to the actual data two days training was given for data collectors by the principal investigator on the data collection tools and the sample collection procedures. The questionnaire was pre-tested on 5% of Gene Xpert–negative presumptive tuberculosis adult patients in Jugal General Hospital to check the validity and the applicability of the questionnaire. All Sputum specimens were collected and processed according to the Standard operating procedure (SOP). The quality of the specimen was checked based on Bartlett's criteria (Popova et al., 2019). The sterility of the culture media was ensured by incubating 5% of each batch of the prepared media at 37°C for 24 hrs. To standardize the inoculum density of bacterial suspension, turbidity was adjusted using 0.5 McFarland standard [27]. The performance of all prepared media was checked by inoculating standard American Type culture collection (ATCC) strains, including *Staphylococcus aureus* (ATCC 25923) and *Escherichia coli* (ATCC 25922), following CLSI guidelines (2022).

## Data analysis

Data were entered into EPI-Data version 4.60 and analyzed using Statistical Package for Social Science (SPSS) program version 20. Results were presented by using text, Table, and figure. Variables with a p-value ≤ 0.25 in bi-variable analysis were selected candidates for multivariable analysis. Finally, the multivariable logistic regression analysis were used for final decisions with odds ratio and 95% confidence intervals. Statistical significance was declared at a p-value < 0.05.

## Ethical consideration

Ethical clearance was obtained from Haramaya University, College of Health and Medical Sciences through the Institutional Health Research Ethics Review Committee (IHRERC) (Ref. No. IHRERC/159/2023). A permission letter to conduct this research was obtained from Hiwot Fana Comprehensive Specialized University Hospital. Study subjects were given information about the objectives of the study and informed written consent or thumbprint for subjects who were unable to write were obtained. Subjects participated voluntarily and could withdraw from the study at any time without further consequences. Study participants who became positive for culture and antimicrobial susceptibilities were linked to antibiotic treatment regime through communicating with clinician.

# Results

## Socio-demographic characteristics of the study participants

From the total of 376 Gene Xpert MTB/RIF-negative adult patients presenting with symptoms of pulmonary TB, 371 were enrolled with a response rate of 98.7%. The proportion of male patients was 197/371 (53.1%). The mean age (±SD) of study participants was 38 ± 15.69. Rural residents accounts 213/371 (57.4%), farmers accounted 129/371 (34.8%), and 146/371 (39.4%) of participants have no formal education (Table 1).

## Clinical, behavioral, and environmental characteristics of the study participants

More than half of study participants 192 (51.8%) had a history of antibiotic usage for a month or more before the study period. Previous hospital admission was reported 40(10.8%) and prior tuberculosis in 17(4.6%). From the total study participants, thirty percent 113(30.5%) of them had the habit of smoking cigarettes, and 155 (41.8%) of them lived in crowded conditions. Majority of them used firewood as a source of energy for cooking 226(60.9%) and half of them 187(50.4%) lived in a mud/soil floor house (Table 2).

## Prevalence of bacterial LRTI pathogens

The overall prevalence of bacterial LRTIs pathogens was 34.0%, (95% CI; 29.2, 38.8). Out of 126 isolated bacterial pathogens, two-third of them were from Gram-negative bacteria (GNB) 84/126(67%) whereas one-third of them corresponded to

Table 1. Socio-demographic characteristics of study participants at HFCSH, Harar, eastern Ethiopia, 2024 (n = 371).

| Variables | Category | Frequency | Percentage (%) |
|---|---|---|---|
| Sex | Male | 197 | 53.1 |
| | Female | 174 | 46.9 |
| Age | 18-35 | 184 | 49.6 |
| | 36-49 | 94 | 25.3 |
| | 50-64 | 66 | 17.8 |
| | Above 65 | 27 | 7.3 |
| Marital status | Unmarried | 119 | 32.1 |
| | Married | 241 | 65.0 |
| | Divorced/separated | 5 | 1.3 |
| | Widowed | 6 | 1.6 |
| Residence | Urban | 158 | 42.6 |
| | Rural | 213 | 57.4 |
| Educational status | No formal Education | 124 | 33.4 |
| | Primary | 146 | 39.4 |
| | Secondary | 58 | 15.6 |
| | College and above | 43 | 11.6 |
| Occupation | Farmer | 127 | 34.2 |
| | Factory | 14 | 3.8 |
| | Government | 49 | 13.2 |
| | Private | 27 | 7.3 |
| | House wife | 68 | 18.3 |
| | Student | 78 | 21.0 |
| | Others | 8 | 2.2 |
| Family size | 2-4 | 145 | 39.1 |
| | 5-7 | 185 | 49.9 |
| | Above 8 | 41 | 11.1 |
| Patient category | Inpatient | 13 | 3.5 |
| | Outpatient | 358 | 96.5 |

Gram-positive bacteria 42/126(33%) Among GNB isolates, *Klebsella species* accounted highest percentage. These includes *K. pneumoniae* 18/126(14.3%), *K. oxytoca* 10/126(7.9%), *K. ozanae* 9/126(7%), and *K. rhinoscleromatis* 13/126(10.3%). The prevalence of other GNB isolates were *E. coli* 8/126(6.3%), *Entrobacter species* 7/126(5.6%), *Citrobacter species* 4/126(3.2%), *Proteus mirabilis* 1/126(0.8%), *H. influenzae* 6/126(4.8%) and *M. catarhallis* 8/126(6.3%). The isolates from Gram-positive bacteria category revealed three isolates namely: *S. aurues* 24/126(19%), *S. pneumonia* 13/126(10.3%), and *S. pyogens* 5/126(4%). *S. aureus* (19.0%) showed the highest predominance among all isolates followed by *K. peumoniae*, (14.3%). *K. rhinoscleromatis* (10.3%) *and S. pneumoniae* (10.3%) *showed* similar rate of distribution which put them in third place (Fig 1). **Percentage of pathogenic bacterial isolates from sputum culture of adult patients with Gene Xpert-negative presumptive tuberculosis at HFCSH, eastern Ethiopia, from January 01 to April 30, 2024.**

## Antimicrobial susceptibility patterns of the bacterial isolates

In this study, the antimicrobial susceptibility patterns of Gram-positive bacteria (n = 42) =isolates produced wider variations in their susceptibility patterns. Overall, gram-positive isolates were susceptible to clindamycin

**Table 2. Clinical history, behavioral, and environmental characteristics of the study participants at HFCSH, Harar, eastern Ethiopia, 2024 (n=371).**

| Variables | Category | Frequency | Percentage (%) |
|---|---|---|---|
| Comorbidity | Diabetes | 12 | 3.2 |
| | HIV | 8 | 2.2 |
| | Heart disease | 7 | 1.9 |
| | Asthma | 7 | 1.9 |
| History of hospital admission | Yes | 40 | 10.8 |
| | No | 331 | 89.2 |
| History of Tuberculosis | Yes | 17 | 4.6 |
| | No | 354 | 95.4 |
| History of antibiotic usage prior a month and above | Yes | 192 | 51.8 |
| | No | 179 | 48.2 |
| Smoking cigarettes | Yes | 113 | 30.5 |
| | No | 258 | 69.5 |
| Alcohol use | Never | 291 | 78.4 |
| | Sometimes | 10 | 2.7 |
| | Always | 70 | 18.9 |
| Chewing khat | Yes | 188 | 50.7 |
| | No | 183 | 49.3 |
| Hand washing after outing | Never | 31 | 8.4 |
| | Sometimes | 307 | 82.7 |
| | Always | 33 | 8.9 |
| Crowded living condition | Yes | 155 | 41.8 |
| | No | 216 | 58.2 |
| Separate kitchen from house | Yes | 335 | 90.3 |
| | No | 36 | 9.7 |
| Source of energy for cooking | Firewood | 226 | 60.9 |
| | Electric power | 145 | 39.1 |
| Living in mud/soil floor | Yes | 187 | 50.4 |
| | No | 184 | 49.6 |
| Exposed to dust at work | Yes | 231 | 62.3 |
| | No | 140 | 37.7 |
| Source of drinking water | Improved | 151 | 40.7 |
| | Unimproved | 220 | 59.3 |

29/42(69.05%) and erythromycin 25/42(59.5%) while 31/42 (73.8%) was resistant to tetracycline Isolates of *S. aureus* were resistant to penicillin 19/24 (79.2%), trimethoprim–sulfamethoxazole 14/24(58.3%) however, 20/25(83.3%) isolates were susceptible to (cefoxitin, ciprofloxacin, and gentamycin), and 18/24(75%) azithromycin (Fig 2). **Antimicrobial susceptibility patterns of S. aureus from sputum culture among adult patients with Gene Xpert-negative presumptive tuberculosis at HFCSH, eastern Ethiopia, from January 01 to April 30, 2024.**

*Among isolates of S. aureus* 4/24 (16.7%) showed zone of inhibition ≤21 mm (16.2 to 19.5 mm)), in the cefoxitin disc diffusion assay and they were identified as methicillin-resistant (MRSA). The isolates of *S. pneumoniae* were resistant to erythromycin 7/13(53.8%), tetracycline 7/13(53.8%), and sensitive to clindamycin 10/13(76.9%), and co-trimoxazole

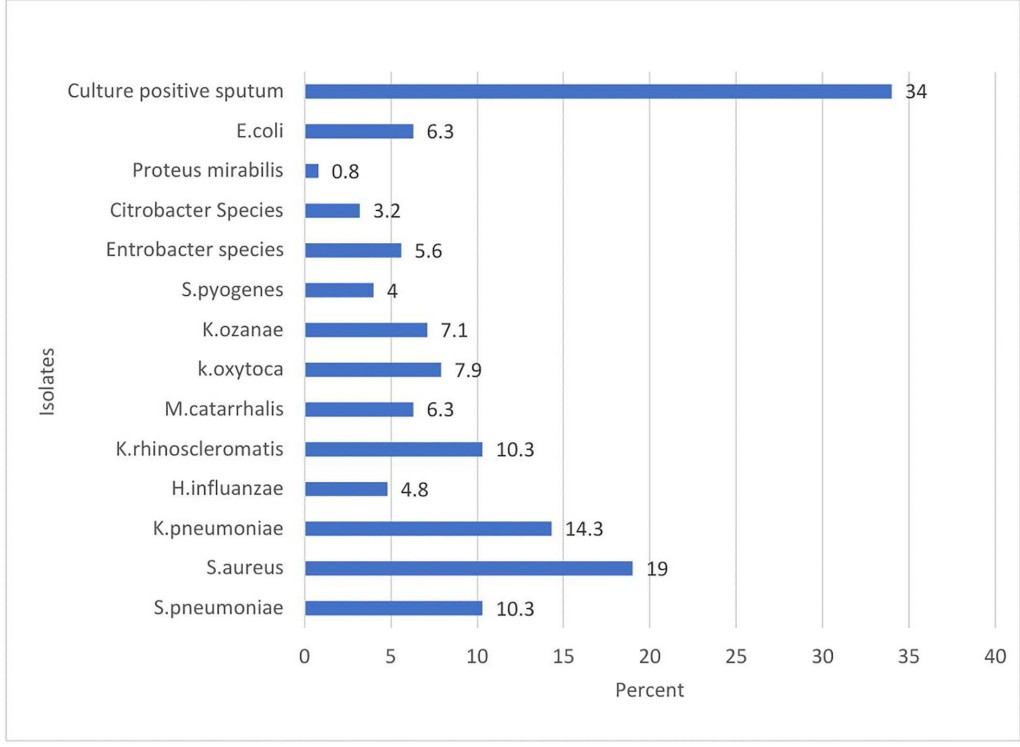

**Fig 1. Percentage of pathogenic bacterial isolates from sputum culture of adult patients with Gene Xpert-negative presumptive tuberculosis at HFCSH, eastern Ethiopia, from January 01 to April 30, 2024.**

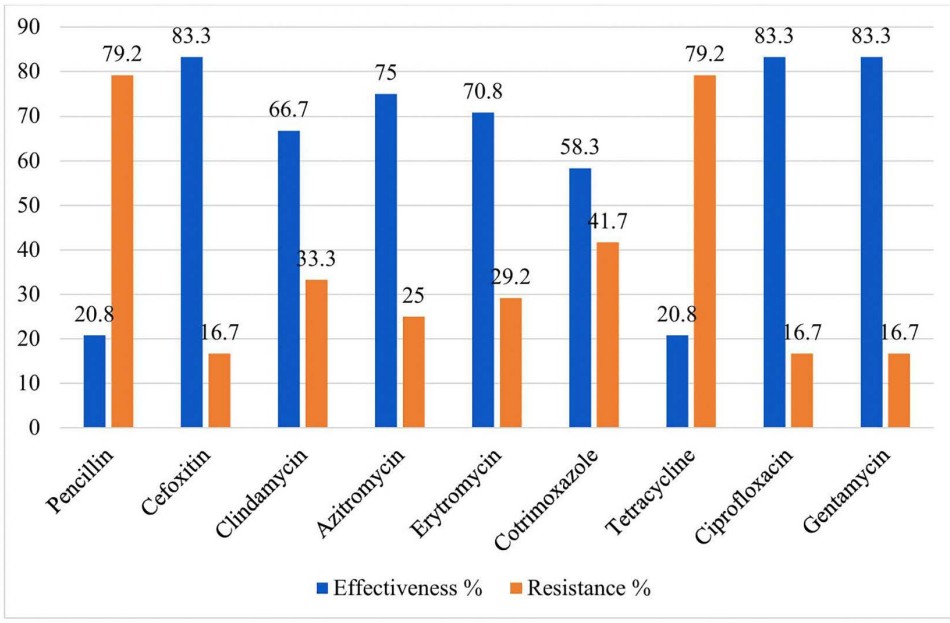

**Fig 2. Antimicrobial susceptibility patterns of S. aureus from sputum culture among adult patients with Gene Xpert-negative presumptive tuberculosis at HFCSH, eastern Ethiopia, from January 01 to April 30, 2024.**

8/13(61.5%). The isolates of *S. pyogenes* were resistant to clindamycin 2/5(40%), erythromycin 3/5(60%), However, all isolates of *S. pyogenes* showed susceptibility towards tetracycline (Table 3).

The antimicrobial Susceptibility patterns of Gram-negative bacteria (n = 84) isolates against nine antibiotics were as follows,almost all of the Gram-negative isolates were susceptible to piperacillin/tazobactum 81/84(96.4%), meropenem 82/84(97.6%); and resistant to ceftriaxone 24/84 (28.6%), and gentamycin 23/78 (29.5%) respectively. The isolates of *K. peumoniae* were found to be resistant to gentamycin, 7/18 (38.9%), amoxicillin-clavulanate (augmentin), 12/18 (66.7%), while all isolates were highly susceptible to piperacillin/ tazobactam, and meropenem. The majority of *E. coli* isolates were resistant to ampicillin, 7/8(88.5%), amoxicillin-clavulanate, 5/8(62.5%) but all isolates were susceptible to piperacillin, and meropenem (Fig 3). **Antimicrobial susceptibility patterns of K. pneumoniea from sputum culture among adult patients with Gene Xpert-negative presumptive tuberculosis at HFCSH, eastern Ethiopia, from January 01 to April 30, 2024.**

A single isolated of *P.mirabilis* was susceptible to all set of used antibiotics. All isolates of *H.influenzae* were resistant to ampicillin but all of them were susceptible to ciprofloxacin, piperacillin, and meropenem. The isolates of *M. catarrhalis* were highly resistant to co-trimoxazole, but all isolates were susceptible to ceftazidime, piperacillin, and meropenem (Table 4).

## Multi-drug resistance patterns of bacterial isolates

Multi-drug resistance (MDR) was found in 50/126 (39.7%). About a third of isolates 16/50 (32%) were resistant to 3 antibiotics and 21/50 (42%) of them were resistant to five and more antibiotics. MDR was accounted 35/84 (41.7%) in Gram-negative bacteria isolates with *E. coli* 6/8 (75%) the predominant MDR species, with followed by *M. catarrhalis* 4/8 (50%), *K. rhinoscleromatis* 6/13 (46.2%) whereas *S. pyogens* 2/5 (40%) were the dominant MDR species among gram positive isolates. The least percentage of MDR was shown in the isolates of *H. influenzae* 1/6 (16.7%) (Table 5).

## Factors associated with bacterial LRTI pathogens

In the bi-variable analysis, gender, educational status, history of hospital admission, history of antibiotic usage prior a month and above, habit of hand washing after outing, crowded living conditions, separate kitchen from house, living in mud floor, and source of drinking water were significantly associated with bacterial LRTI pathogens with

**Table 3. Antimicrobial susceptibility patterns of Gram-positive bacterial isolates from sputum culture of adult patients with Gene Xpert -negative presumptive tuberculosis at HFCSH, eastern Ethiopia, 2024 (N = 42).**

| Bacterial Isolates | Total N (%) | Pattern | Antimicrobial agent's n (%) | | | | | | | | |
|---|---|---|---|---|---|---|---|---|---|---|---|
| | | | P | FOX | CD | AZT | E | SXT | TTC | CIP | CN |
| *S. aureus* | 24 (19) | S | 5 (20.8) | 20 (83.3) | 16 (66.7) | 18 (75) | 17 (70.8) | 14 (58.3) | 5 (20.8) | 20 (83.3) | 20 (83.3) |
| | | R | 19 (79.2) | 4 (16.7) | 8 (33.3) | 6 (25) | 7 (29.2) | 10 (41.7) | 19 (79.2) | 4 (16.7) | 4 (16.7) |
| *S. pneumoniae* | 13 (10.3) | S | NA | NA | 10 (76.9) | NA | 6 (46.2) | 8 (61.5) | 6 (46.2) | NA | NA |
| | | R | | | 3 (23.1) | | 7 (53.8) | 5 (38.5) | 7 (53.8) | | |
| *S. pyogenes* | 5 (4) | S | NA | NA | 3 (60) | NA | 2 (40) | NA | 0 | NA | NA |
| | | R | | | 2 (40) | | 3 (60) | | 5(100) | | |
| Total | 42 (33.3) | S | 5 (20.8) | 20 (83.3) | 29 (69) | 18 (75) | 25 (59.5) | 22 (59.5) | 11 (26.2) | 20 (83.3) | 20 (83.3) |
| | | R | 19 (79.2) | 4 (16.7) | 13 (31) | 6 (25) | 17 (40.5) | 15 (40.5) | 31 (73.8) | 4 (16.7) | 4 (16.7) |

Note: S = Sensitive, R = Resistant, NA = Not applicable, (P = Penicillin, FOX = Cefoxitin, CD = Clindamycin, AZT = Azithromycin, E = Erythromycin, SXT = tri-methoprim–sulfamethoxazole, TTC = Tetracycline, CIP = Ciprofloxacin, CN = Gentamycin).

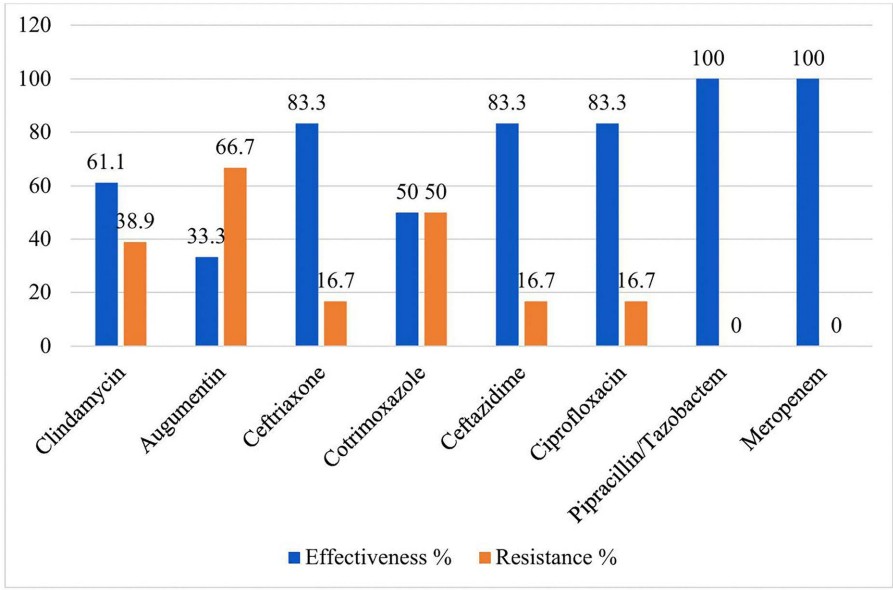

**Fig 3. Antimicrobial susceptibility patterns of K. pneumoniea from sputum culture among adult patients with Gene Xpert-negative presumptive tuberculosis at HFCSH, eastern Ethiopia, from January 01 to April 30, 2024.**

*p*-value less than 0.25. However, in the multivariable logistic regression analysis model, educational status, history of prior antibiotic usage and crowded living conditions remained associated with bacterial LRTI pathogens at *p*-value less than 0.05.

Study participants who had primary educational status were four times (AOR = 4.01; 95% CI = 1.62, 9.92; *p* = 0.003) more likely to be infected with bacterial LRTI pathogens compared to those who had college and above educational status. Study participants who had a history of antibiotic usage prior a month and above) were two almost times (AOR = 1.87; 95% CI = 1.13, 3.08; *p* = 0.014) more likely to be infected with bacterial LRTI pathogens compared to their counterparts. Study participants who were living in a crowded conditions were five times (AOR = 5.11; 95% CI = 3.03, 8.61; *p* < 0.001) more likely to be infected with bacterial LRTI pathogens compared to counterparts (Table 6).

## Discussion

Lower respiratory tract infections (LRIs) are a significant public health problem and a leading cause of morbidity and mortality in persons of any age [16]. In the current study, the overall prevalence of bacterial LRTIs among Gene Xpert-negative presumptive pulmonary tuberculosis adult patients was found to be 34.0% (95% CI; 29.2, 38.8). Comparable findings were reported from previous studies conducted in Hawassa (33.5%) [21], and Addis Ababa, Ethiopia [22]. However, this prevalence was lower than the study conducted in couple of cities in Ethiopia Gondar (57.1%) [5] and Jimma (80%) [20], Cameroon (46.8%) [30], Cambodia (44%) [31], Nepal (47.33%) [26], Philippines (90.24%) [32]. This discrepancy might be due to variation in diagnosis methods used by those studies that increased detection capacity including detecting atypical bacteria that could not be detected through routine culture. They used molecular test like commercial reverse-transcriptase real-time polymerase chain reaction (RT-PCR) assay and 16S rRNA gene amplicon sequencing which results in an increasing the detection rate of all bacteria than conventional culture method we used [30,32,33]. However, this study is higher than study conducted in Tanzania (16.4%) [18]. This variation could be due to geographical variation, difference in study population and sample size [34].

**Table 4. Antimicrobial susceptibility patterns of Gram-negative bacterial isolates from sputum culture of adult patients with Gene Xpert MTB/ RIF-negative presumptive tuberculosis at HFCSH, eastern Ethiopia,2024 (N = 84).**

| Bacterial species | N(%) | Pattern | Antimicrobial agents | | | | | | | | |
|---|---|---|---|---|---|---|---|---|---|---|---|
| | | | AMP | CN | AUG | CRO | SXT | CAZ | CIP | TZP | MEM |
| *K. pneumoniae* | 18 (14.3) | S | NA | 11 (61.1) | 6 (33.3) | 15 (83.3) | 9 (50) | 15 (83.3) | 15 (83.3) | 18 (100) | 18 (100) |
| | | R | | 7 (38.9) | 12 (66.7) | 3 (16.7) | 9 (50) | 3 (16.7) | 3 (16.7) | 0 | 0 |
| *K. oxytoca* | 10 (7.9) | S | NA | 8 (80) | 3 (30) | 6 (60) | 7 (70) | 7 (70) | 9 (90) | 10 (100) | 9 (90) |
| | | R | | 2 (20) | 7 (70) | 4 (40) | 3 (10) | 3 (10) | 1 (10) | 0 | 1 (10) |
| *K. ozanae* | 9(7.1) | S | NA | 7 (77.8) | 4 (44.4) | 5 (55.6) | 5 (55.6) | 6 (66.7) | 7 (77.8) | 8 (88.9) | 8 (88.9) |
| | | R | | 2 (22.2) | 5 (55.6) | 4 (44.4) | 4 (44.4) | 3 (33.3) | 2 (22.2) | 1 (11.1) | 1 (11.1) |
| *K. rhinoscleromatis* | 13(10.3) | S | NA | 8 (61.5) | 6 (46.2) | 9 (69.2) | 6 (46.2) | 8 (61.5) | 8 (61.5) | 11 (84.6) | 13 (100) |
| | | R | | 5 (38.5) | 7 (53.8) | 4 (30.8) | 7 (53.8) | 5 (38.5) | 5 (38.5) | 2 (15.2) | 0 |
| *E. coli* | 8 (6.3) | S | 1 (12.5) | 7 (87.5) | 3 (37.5) | 5 (62.5) | 5 (62.5) | 5 (62.5) | 4 (50) | 8 (100) | 8 (100) |
| | | R | 7(88.5) | 1 (12.5) | 5 (62.5) | 3 (37.5) | 3 (37.5) | 3 (37.5) | 4 (50) | 0 | 0 |
| *Enterobacter spp.* | 7 (5.6) | S | NA | 4 (57.1) | 4 (57.1) | 5 (71.4) | 4 (57.1) | 7 (100) | 6 (85.7) | 7 (100) | 7 (100) |
| | | R | | 3 (42.9) | 3 (42.9) | 2 (28.6) | 3 (42.9) | 0 | 1 (14.3) | 0 | 0 |
| *Citrobacter spp.* | 4 (3.2) | S | NA | 4 (100) | NA | 3 (75) | 3 (75) | 1 (25) | 4 (100) | 4 (100) | 4 (100) |
| | | R | | 0 | | 1 (25) | 1 (25) | 3 (75) | 0 | 0 | 0 |
| *P. mirabilis* | 1(0.8) | S | 1 (100) | 1 (100) | 1 (100) | 1 (100) | 1 (100) | 1 (100) | 1 (100) | 1 (100) | 1 (100) |
| | | R | 0 | 0 | 0 | 0 | 0 | 0 | 0 | 0 | 0 |
| *H. influenzae* | 6 (4.8) | S | 0 | NA | 6 (100) | 5 (83.3) | 5 (83.3) | NA | 6 (100) | 6 (100) | 6 (100) |
| | | R | 6(100) | | 0 | 1 (16.7) | 1 (16.7) | | 0 | 0 | 0 |
| *M. catarrhalis* | 8 (6.3) | S | 4 (50) | 5(62.5) | 4(50) | 6(75) | 1(12.5) | 8 (100) | 6 (75) | 8 (100) | 8 (100) |
| | | R | 4 (50) | 3(37.5) | 4(50) | 2(25) | 7(87.5) | 0 | 2 (25) | 0 | 0 |
| Total | 84 (66.7) | S | - | 55 (70.5) | - | 60 (71.4) | 46 (54.8) | - | 66 (78.6) | 81 (96.4) | 82 (97.6) |
| | | R | | 23 (29.5) | | 24(28.6) | 38 (45.2) | | 18 (21.4) | 3 (3.6) | 2 (2.4) |

Note: S = Sensitive, R = Resistant, NA = Not applicable, (AMP = Ampicillin, CN = Gentamycin, AUG = Augmentin, CRO = Ceftriaxone, SXT = trimethoprim–sulfamethoxazole, CAZ = Ceftazidime, CIP = Ciprofloxacin, TZP = Pipracillin/Tazobactem, MEM = Meropenem)

**Table 5. Multi-drug resistance patterns of bacterial isolates from sputum culture of adult patients with Gene Xpert-negative presumptive tuberculosis at HFCSH, eastern Ethiopia, 2024 (N = 50).**

| Bacterial isolates | Level of resistance pattern n (%) | | | | | | |
|---|---|---|---|---|---|---|---|
| | RO | R1 | R2 | R3 | R4 | ≥ R5 | Total MDR ≥ R3 |
| *S. aureus* (n = 24) | 0 | 0 | 0 | 0 | 3(12.5) | 6(25) | 9(37.5) |
| *S. pneumoniae* (n = 13) | 0 | 0 | 0 | 1(7.7) | 3(23.1) | 0 | 4(30.7) |
| *S. pyogenes* (n = 5) | 0 | 0 | 1(20) | 1(20) | 0 | 1(20) | 2(40) |
| *K. pneumoniae* (n = 10) | 2(11.1) | 0 | 0 | 3(16.7) | 0 | 3(16.7) | 6(33.3) |
| *K. oxytoca* (n = 9) | 1(10) | 2(20) | 1(10) | 2(20) | 1(10) | 1(10) | 4(40) |
| *K. ozanae* (n = 13) | 0 | 2(22.2) | 2(22.2) | 1(11.1) | 2(22.2) | 1(11.1) | 4(44.4) |
| *K. rhinoscleromatis* (n = 8) | 1(7.7) | – | 1(7.7) | – | 1(7.7) | 5(38.5) | 6(46.2) |
| *E. coli* (n = 7) | 2(25) | 1(12.5) | 0 | 3(37.5) | 1(12.5) | 2(25) | 6(75) |
| *Enterobacter species* (n = 4) | 3(42.8) | 1(14.3) | 1(14.3) | 3(42.8) | 0 | 0 | 3(42.8) |
| *Citrobacter Species* (n = 1) | 4(100) | 2(50) | 0 | 1(25) | 0 | 0 | 1(25) |
| *Proteus mirabilis* (n = 6) | 1(100) | 0 | 0 | 0 | 0 | 0 | 0 |
| *H. influenzae* (n = 8) | 4(66.7) | 2(33.3) | 0 | 0 | 0 | 1(16.7) | 1(16.7) |
| *M. catarrhalis* (n = 13) | 3(37.5) | 0 | 2(25) | 1(12.5) | 2(25) | 1(12.5) | 4(50) |
| Total (n = 126) | 21(16.7) | 10(7.9) | 8(6.3) | 16(12.6) | 13(10.3) | 21(16.7) | 50(39.6) |

**Table 6. Bi-variable and multivariable logistic regression analyses of factors associated with bacterial LRTI pathogens among Gene Xpert-negative presumptive tuberculosis adult patients at HFCSH, eastern Ethiopia, 2024.**

| Variables | Culture result | | Bi-variable analysis | Multivariable analysis | |
|---|---|---|---|---|---|
| | Positive n (%) | Negative n (%) | COR (95% CI) | AOR (95% CI) | *p*-value |
| Gender | | | | | |
| Male | 60(30.5) | 137(69.5) | 1 | | |
| Female | 66(37.9) | 108(62.1) | 1.39(.91, 2.15) | | |
| Age | | | | | |
| 18-35 | 63(34.2) | 121(65.8) | 1 | | |
| 36-49 | 28(29.8) | 66(70.2) | 0.82(0.47,1.39) | | |
| 50-64 | 23(34.8) | 43(65.2) | 1.03(0.56,1.84) | | |
| 65 and above | 12(44.4) | 15(55.6) | 1.54(0.67, 3.48) | | |
| Marital status | | | | | |
| Unmarried | 39(32.8) | 80(67.2) | 0.98(0.18,7.25) | | |
| Married | 84(34.9) | 157(65.1) | 1.07(0.21,7.8) | | |
| Divorce | 1(20) | 4(80) | 0.50(0.02,7.54) | | |
| Widowed | 2(33.3) | 4(66.7) | 1 | . | |
| Residence | | | | | |
| Urban | 53(33.5) | 105(66.5) | 1 | | |
| Rural | 73(34.3) | 140(65.7) | 1.03(0.67,1.59) | | |
| Educational level | | | | | |
| No informal | 45(36.3) | 79(63.7) | 0.87(0.43,1.78) | | |
| Primary | 44(30.1) | 102(69.9) | 0.66(0.33,1.35) | **4.01(1.62,9.92)** | **0.003**\** |
| Secondary | 20(34.5) | 38(65.5) | 0.81(0.36,1.83) | | |
| Tertiary | 17(39.5) | 26(60.5) | 1 | | |
| Occupational status | | | | | |
| Farmer | 35(27.6) | 92(72.4) | 0.63(0.15,3.20) | | |
| Factory | 5(35.7) | 9(64.3) | 0.93(0.15,6.09) | | |
| Government | 18(36.7) | 31(63.3) | 0.97(0.21,5.16) | | |
| Private | 11(40.7) | 16(59.3) | 1.15(0.23,6.53) | | |
| House wife | 27(39.7) | 41(60.3) | 1.2(0.25,5.70) | | |
| Student | 27(34.6) | 51(65.4) | 0.882(0.201,4.562) | | |
| Other | 3(37.5) | 5 (62.5) | 1 | | |
| Family size | | | | | |
| 2-4 | 45(31) | 100(69) | 1 | | |
| 5-7 | 67(36.2) | 118(63.8) | 1.26(0.79,2.01) | | |
| Above 8 | 14(34.1) | 27(65.9) | 1.152(0.54,2.38) | | |
| Alcohol use | | | | | |
| Never | 94(32.3) | 197(67.7) | 1 | | |
| Sometimes | 5(50) | 5(50) | 2.09(0.57,7.70) | | |
| Always | 27(38.6) | 43(61.4) | 1.32(0.76,2.25) | | |
| Smoking | | | | | |
| Yes | 43(38.1) | 70(61.9) | 1.29(0.81,2.05) | | |
| No | 83(32.2) | 175(67.8) | 1 | | |
| Chewing khat | | | | | |
| Yes | 62(33) | 126(67) | 0.92(0.59,1.41) | | |
| No | 64(35) | 119(65) | 1 | | |

*(Continued)*

| Variables | Culture result | | Bi-variable analysis | Multivariable analysis | |
|---|---|---|---|---|---|
| | Positive n (%) | Negative n (%) | COR (95% CI) | AOR (95% CI) | p -value |
| Comorbidity | | | | | |
| DM | 5(41.7) | 7(58.3) | 1.79(0.26,1.62) | | |
| HIV | 3(37.5) | 5(62.5) | 1.50(0.170,15.55) | | |
| HD | 2(28.6) | 5(71.4) | 1.00(0.09,11.32) | | |
| Asthma | 2(28.6) | 5(71.4) | 1 | | |
| Admission history | | | | | |
| Yes | 19(47.5) | 21(52.5) | 1.89(0.97,3.68) | | |
| No | 107(32.3) | 224(67.7) | 1 | | |
| History of TB | | | | | |
| Yes | 6(35.3) | 11(64.7) | 1.06(0.36,2.87) | | |
| No | 120(33.9) | 234(66.1) | 1 | | |
| Prior antibiotic usage | | | | | |
| Yes | 81(42.2) | 111(57.8) | 2.17(1.40, 3.40) | **1.86(1.13,3.08)** | **0.014*** |
| No | 45(25.1) | 134(74.9) | 1 | | |
| Hand washing | | | | | |
| Never | 15(48.4) | 16(51.6) | 0.99 (0.37,2.67) | | |
| Sometimes | 95(30.9) | 212(69.1) | 0.48(0.23, 0.99) | | |
| Always | 16(48.5) | 17(51.5) | 1 | | |
| Crowded living | | | | | |
| Yes | 86(55.5) | 69(44.5) | 5.48(3.46, 8.82) | **5.10 (3.03, 8.61)** | **<0.001*** |
| No | 40(18.5) | 176(81.5) | 1 | | |
| Separate kitchen | | | | | |
| Yes | 119(35.5) | 216(64.5) | 1 | | |
| No | 7(19.4) | 29(80.6) | 0.44(0.17,0.98) | | |
| Cooking energy | | | | | |
| Firewood | 75(33.2) | 151(66.8) | 0.92(0.59,1.42) | | |
| Electric | 51(35.2) | 94(64.8) | 1 | | |
| Dust exposure | | | | | |
| Yes | 75(32.5) | 156(67.5) | 0.84(0.54,1.31) | | |
| No | 51(36.4) | 89(63.6) | 1 | | |
| Drinking water source | | | | | |
| Unimproved | 66(30) | 154(70) | 0.65(0.42,1.01) | | |
| Improved | 60(39.7) | 91(60.3) | 1 | | |

All culture positive rate of bacteriological confirmed cases were found to be mono-bacterial infections. This study is consistent with the study reported from Ghana [35]. However, it is different from the study done in different parts of Ethiopia, Addis Ababa [22], Gondar [5], Hawassa [21], Tanzania [18], Iran [36] and China [37]. The difference might be due to the difference in used empirical treatment, variation of diagnosis method, and sample size.

The most common pathogenic bacteria isolates were *S. aureus, S. pneumoniae, S. pyogenes, K. penumoniae, K. oxyzanae, K.ozanae, K. rhinoscleromatis, Enterobacter species, Citrobacter species, E. coli, H. influnzae, M. catarhallis, and P. mirabillis*. This is concordance with the study done in Gondar, Ethiopia [5] and Tanzania [18]. However, this study was inconsistent with studies done in Nepal [26] and Philippines [32]. This might be due to that the etiologic agents may

vary with epidemiological variation [12,30]. The two-third of bacterial isolates were belongs to gram-negative bacteria 84 (66.7%). The same data were reported by study done in several parts of Ethiopia,Addis Ababa (77.8%) [22], Gondar (64.9%) [5], and Hawassa (76.1%) [21] and other countries like Tanzania (75%) [18] and China (72.6%) [37]. This might be due to that GNB have strong pathogenic ability and virulence factors such as Type1 pili, which may enhance the ability to adhere and colonize the lower respiratory tract and outer membrane that aid in pathogenesis and immune evasions [38]. However, in contrast to studies reported in Iran [36], Nigeria [39], and Jimma, Ethiopia [40] Gram-positive bacteria were the dominant group. S. aureus 24 (19%) was the most dominant isolates among all. Similar dominance of S. aureus isolates was noted in the studies carried out in different places, Gondar Ethiopia [5], and in Pakistan [41].

Furthermore, S. aureus persistence in respiratory tract is supported by a variety of traits including demonstrating metabolic versatility, its capacity to scavenge iron, coordinated regulation of gene expression, horizontal genes transfer, and expression of surface adhesions [42].

K. pneumoniae (14.3%) were the second most frequently isolated bacteria. This is in line with study carried out in Gondar, Ethiopia [5] and Tanzania [18]. Most of Gram-positive bacteria (83%) including S. aureus isolates were highly susceptible to ciprofloxacin, gentamicin, and cefoxitin while showed resistance to penicillin (79.2%) and tetracycline (73.8%). These observations are in line with several studies done across different parts of Ethiopia [5,21,22,37]. It is to be noted that almost all of the isolates of Gram-negative bacteria were highly susceptible to piperacillin/tazobactam (96.4%) and meropenem (97.6%) respectively while exhibiting resistance to amoxicillin-clavulanate. This is in line with the study done in Addis Ababa, Ethiopia [22]. All isolates of K. pneumoniae were susceptible to piperacillin/tazobactam (100%) and meropenem (100% and on the contrary half of them were resistant to trimethoprim–sulfamethoxazole (50%), whereas two-third of them were resistant to amoxicillin-clavulanate (66.7%). These are comparable to the studies done in a couple of cities of Ethiopia [21,22] and China [37].

In this study, MDR was observed in the case of 50 (39.7%) of isolates, which is comparable to a study conducted in Hawassa, Ethiopia (32.4%) [21]. However, lower than the studies done in Addis Ababa, Ethiopia (80.2%) [22] and Nepal (55.56%) [26]. The difference of antibiotic resistance may be owing to wide spread irrational use and over exploitation of antimicrobial agents [43]. Among the MDR, Majority were corresponding to Gram-negative bacteria 35 (41.7%). These findings are in line with the results of a study conducted in Nepal [26]. The might be due to GNB increasing the intrinsic resistance (mutations) in chromosomal genes such as increasing the expression of antibiotic-inactivating enzymes, efflux pumps, permeability or target modifications acquired by transfer of mobile genetic elements carrying resistance genes such as plasmid [44].

In this study, the methicillin resistant S. aureus (MRSA) strains were found to be 4 (16.7%). A similar pattern of resistance was observed in a study done in Tanzania [18]. This might be due to that MRSA circulating in health facilities and communities from frequent use of broad antibiotics leads to spread of strain that produces beta-lactamase, and through expression of mecA gene that encode production of an unusual penicillin-binding protein and modification of penicillin-binding protein results in weakening the affinity for β-lactam antibiotics [45,46].

Among the various factors assessed contributing to bacterial LRTI pathogens, primary education, history of prior antibiotic usage and crowded living conditions were found to be significantly associated with bacterial LRTI pathogens. Study participants who had primary educational status were 4 times (AOR = 4.01; 95% CI = 1.62, 9.92; p = 0.003) more likely to be infected with bacterial LRTI pathogens compared to those who had college and above educational status. It is in line with studies done in Wolaita Sodo, Ethiopia [47] and China [48]. This may be due to the fact that low education could be a good proxy for conditions of vulnerability because of lack of awareness, but on another way high levels of education would allow us a better understanding about preventive measures such as vaccination [49]. Study participants who had a history of antibiotic usage prior a month and above were almost 2 times (AOR = 1.87; 95% CI = 1.13, 3.01; p = 0.014) more likely to be infected with bacterial LRTI pathogens compared to counterparts. This is consistent with a couple of studies conducted in China [37,50] and USA [51]. The probable reason for this could be due to antibiotic resistance in the bacteria following

the frequent exposure to antibiotics [43,52]. Study participants who were living in a crowded condition were 5 times (AOR = 5.11; 95% CI = 3.03, 8.61; $p < 0.001$) more likely to be infected with bacterial LRTI pathogens compared to their counterparts. The similar patterns showed in a couple of studies done in Gondar and Felege Hiwot, Ethiopia [25,53]. This might be due to the fact that crowdedness facilitates the proximity of individuals to each other that increase the possibility of aerosol transmission and easily accessible infection [24].

### Study limitations

This study did not identified definitive atypical bacterial pathogens associated with LRTIs such as *Chlamydia*, *Mycoplasma*, *Legionella species* and anaerobic bacteria. We acknowledge that, this limitation may results in an underestimation of the actual burden of bacterial LRTIs, as our finding relied on routine culture and biochemical identification method instead of molecular methods such as PCR or 16S rRNA sequencing which could detect a broader range of pathogen.

Furthermore, Extended Spectrum Beta-lactamase (ESBL) and Carbapenemase were not detected. Moreover, this study did not categorize the distribution of bacterial LRTI pathogens as community acquired or hospital acquired.

### Conclusion

This study highlights critical public health issue that should not be overlooked. The current study revealed that more than one-third of patients with presumptive pulmonary tuberculosis adult patients who tested Gene Xpert-negative were found to be infected with pathogenic bacteria. Alarmingly, *S aureus* is emerged as a leading isolate followed by *K. pneumoniae.* In addition, this finding revealed high rates of MDR and MRSA among the identified isolates. Therefore, prioritizing sputum culture among presumptive tuberculosis patient found negative with Gene Xpert, and antimicrobial susceptibility testing is recommended toto identify the causative agents and tailor treatment strategies.

Furthermore, performing molecular methods (such as PCR or 16S rRNA sequencing) if applicable could indicate the true burden of LRTIs rather than relying on routine culture and biochemical identification. Finally, promoting health education, combating antibiotics misuse and improving living conditions could mitigate bacterial LRTIs and safeguard public from this overlooked threats.

### Supporting information

**S1 Data. SPSS raw data supporting the study.**
(SAV)

### Acknowledgments

We would like to thank Haramaya University, College of Health and Medical Science for giving this opportunity to conduct this research. We would also be grateful to data collectors for their commitment and study participants for their participation in this study.

### Author contributions

**Conceptualization:** Negesso Gebre, Kedir Urgesa, Mohammed Ahmed.

**Data curation:** Negesso Gebre, Kedir Urgesa, Mohammed Ahmed.

**Formal analysis:** Negesso Gebre, Kedir Urgesa, Jemal Mohammed, Mohammed Ahmed.

**Funding acquisition:** Negesso Gebre.

**Investigation:** Negesso Gebre.

**Methodology:** Negesso Gebre, Kedir Urgesa, Jemal Mohammed, Rudwan Yasin Abrahim, Mohammed Ahmed.

**Project administration:** Negesso Gebre, Mohammed Ahmed.

**Resources:** Negesso Gebre.

**Software:** Negesso Gebre, Kedir Urgesa, Jemal Mohammed.

**Supervision:** Kedir Urgesa, Jemal Mohammed, Rudwan Yasin Abrahim, Mohammed Ahmed.

**Validation:** Kedir Urgesa, Jemal Mohammed, Rudwan Yasin Abrahim, Mohammed Ahmed.

**Visualization:** Kedir Urgesa, Jemal Mohammed, Rudwan Yasin Abrahim, Mohammed Ahmed.

**Writing – original draft:** Negesso Gebre, Kedir Urgesa, Jemal Mohammed, Rudwan Yasin Abrahim, Mohammed Ahmed.

**Writing – review & editing:** Kedir Urgesa, Jemal Mohammed, Rudwan Yasin Abrahim, Mohammed Ahmed.

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
