## [Decision Letter · Decision Letter 0]

28 Jan 2026

PONE-D-25-64253Uncovering Overlooked Bacterial Infections in GeneXpert MTB/RIF-Negative Sputum Samples of adult patients with presumptive tuberculosis in Harar, Eastern EthiopiaPLOS One

Dear Dr. Ahmed,

Thank you for submitting your manuscript to PLOS ONE. After careful consideration, we feel that it has merit but does not fully meet PLOS ONE’s publication criteria as it currently stands. Therefore, we invite you to submit a revised version of the manuscript that addresses the points raised during the review process.

Please submit your revised manuscript by Mar 14 2026 11:59PM. If you will need significantly more time to complete your revisions, please reply to this message or contact the journal office at plosone@plos.org. Please include the following items when submitting your revised manuscript:

We look forward to receiving your revised manuscript.

Kind regards,

Frederick Quinn

Academic Editor

PLOS One

3. In the online submission form you indicate that your data is not available for proprietary reasons and have provided a contact point for accessing this data. Please note that your current contact point is a co-author on this manuscript. According to our Data Policy, the contact point must not be an author on the manuscript and must be an institutional contact, ideally not an individual. Please revise your data statement to a non-author institutional point of contact, such as a data access or ethics committee, and send this to us via return email. Please also include contact information for the third party organization, and please include the full citation of where the data can be found.

4. Please upload a copy of Figure 1, to which you refer in your text on page 10. If the figure is no longer to be included as part of the submission please remove all reference to it within the text.

5. Please include your tables as part of your main manuscript and remove the individual files. Please note that supplementary tables (should remain/ be uploaded) as separate "supporting information" files

Reviewers' comments:

Reviewer's Responses to Questions

**Comments to the Author**

1. Is the manuscript technically sound, and do the data support the conclusions?

Reviewer #1: Yes

Reviewer #2: Yes

2. Has the statistical analysis been performed appropriately and rigorously? 

Reviewer #1: Yes

Reviewer #2: N/A

3. Have the authors made all data underlying the findings in their manuscript fully available?

Reviewer #1: No

Reviewer #2: Yes

4. Is the manuscript presented in an intelligible fashion and written in standard English?

Reviewer #1: Yes

Reviewer #2: No

5. Review Comments to the Author

Reviewer #1: This study investigates the bacterial profile and antibiotic resistance patterns in adult patients who present with symptoms of pulmonary tuberculosis (TB) but test negative via the GeneXpert MTB/RIF assay in Harar, Eastern Ethiopia. Using a cross-sectional design with 371 participants, the authors found that 34.0% of these "TB-negative" presumptive cases were suffering from non-tuberculous bacterial infections. The study identifies Gram-negative pathogens, specifically K. pneumoniae, and Gram-positive S. aureus as the predominant isolates. Furthermore, the findings highlight a significant prevalence of multi-drug resistance (39.7%) and identify key risk factors, including crowded living conditions, educational status, and prior antibiotic usage.

The manuscript addresses a critical clinical challenge in many high TB-burden regions. Patients who test negative for TB are often sent home with general antibiotics or left without a definitive diagnosis. This study provides valuable regional data that could significantly improve diagnostic algorithms and antibiotic stewardship in Eastern Ethiopia. The methodology appears technically sound and generally adheres to established microbiological standards. However, the manuscript need clarification on following points.

Comments to authors:

Introduction

Page 11, Lines 92-95: The authors state that studies on this topic are "rare". To ensure scientific precision, please clarify if 'rare' refers to a national context (Ethiopia) or specifically to the Eastern region (Harar). The authors are encouraged to acknowledge and cite relevant existing work to better contextualize their study. For example -

a) Yohannes, Hana, et al. "Pathogenic bacteria recovered from Gene X-pert tuberculosis-negative adult patients in Gondar, Northwest Ethiopia." BMC pulmonary medicine 23.1 (2023): 197.

b) Kebede, Wakjira, et al. "Bacterial pathogens in Xpert MTB/RIF Ultra-negative sputum samples of patients with presumptive tuberculosis in a high TB burden setting: a 16S rRNA analysis." Microbiology Spectrum 12.2 (2024): e02931-23.

Methods and Results

Figure 1 missing (Page 16, Line 234): The manuscript mentions "Figure 1," but this figure appears to be missing from the submitted files and supplementary data. Please provide this figure for review.

To improve the clarity and impact of the results, the authors are encouraged to convert some of the dense tabular data into figure/plots, while keeping the raw data in table/supplementary table:

1) Table 6: This table would be better represented as a Forest Plot. Visualizing the Adjusted Odds Ratios (AOR) and 95% Confidence Intervals would allow for an immediate comparison of the significance of risk factors like crowding and education.

2) Tables 3 & 4: The antimicrobial susceptibility data is quite dense. The authors should consider using grouped or stacked bar charts to represent resistance patterns for the most prevalent isolates (e.g., S. aureus and K. pneumoniae). This would clearly highlight the high resistance to first-line drugs like Ampicillin versus the susceptibility to Meropenem.

Discussion

The study found that 34% of patients had a bacterial infection. This means 66% of the symptomatic patients remained undiagnosed. The authors should expand the discussion to address this "gap." Is it possible that viral pathogens, fastidious bacteria not captured by standard culture, or Non-Tuberculous Mycobacteria (NTM) are responsible for these symptoms?

Typically, S. pneumoniae is the leading cause of community-acquired LRTIs. However, this study found S. aureus (19%) to be the most common in the tested participants. The authors should discuss why Harar might have such high S. aureus levels.

Limitations

The authors should acknowledge in the limitations section that the reliance on manual biochemical identification rather than molecular methods (such as PCR or 16S rRNA sequencing) likely resulted in an underestimation of the true bacterial burden, particularly for fastidious or non-culturable organisms.

Reviewer #2: This articles needs to be modified according to the suggestions made in the comments of the manuscript.

The manuscript does not mention about the study being approved by the institutional research committee along with approval number to be mentioned.

Materials and methods some of the information are added which are not necessary to be included in the manuscript

Conclusion must be written briefly

6. PLOS authors have the option to publish the peer review history of their article (what does this mean?). If published, this will include your full peer review and any attached files.

Reviewer #1: **Yes:** Rajendra K Angara

Reviewer #2: No

---

## [Author Response · Author response to Decision Letter 1]

10 Apr 2026

The authors have attached the response for the editor and reviewer under the file attachment window.

---

## [Decision Letter · Decision Letter 1]

24 Apr 2026

Uncovering Overlooked Bacterial Infections in GeneXpert MTB/RIF-Negative Sputum Samples of adult patients with presumptive tuberculosis in Harar, Eastern Ethiopia

PONE-D-25-64253R1

Dear Dr. Ahmed,

We’re pleased to inform you that your manuscript has been judged scientifically suitable for publication and will be formally accepted for publication once it meets all outstanding technical requirements.

Kind regards,

Frederick Quinn

Academic Editor

PLOS One

Additional Editor Comments (optional):

Reviewers' comments:

Reviewer's Responses to Questions

**Comments to the Author**

1. If the authors have adequately addressed your comments raised in a previous round of review and you feel that this manuscript is now acceptable for publication, you may indicate that here to bypass the “Comments to the Author” section, enter your conflict of interest statement in the “Confidential to Editor” section, and submit your "Accept" recommendation.

Reviewer #1: All comments have been addressed

2. Is the manuscript technically sound, and do the data support the conclusions?

Reviewer #1: Yes

3. Has the statistical analysis been performed appropriately and rigorously? 

Reviewer #1: Yes

4. Have the authors made all data underlying the findings in their manuscript fully available?

Reviewer #1: Yes

5. Is the manuscript presented in an intelligible fashion and written in standard English?

Reviewer #1: Yes

6. Review Comments to the Author

Reviewer #1: All my previous concerns have been satisfactorily addressed in the revised manuscript. The revisions have improved the clarity and quality of the paper.

7. PLOS authors have the option to publish the peer review history of their article (what does this mean?). If published, this will include your full peer review and any attached files.

Reviewer #1: No

---

## [Editor Report · Acceptance letter]

PONE-D-25-64253R1

PLOS One

Dear Dr. Ahmed,

I'm pleased to inform you that your manuscript has been deemed suitable for publication in PLOS One. Congratulations! Your manuscript is now being handed over to our production team.

Kind regards,

on behalf of

Dr. Frederick Quinn

Academic Editor

PLOS One